# Using Interdisciplinary Techniques for Digital Reconstruction of Anti-Turkish Fortification Watchtower

Rok Kamnik [1,*], Saša Djura Jelenko [2], Matjaž Nekrep Perc [1] and Marko Jaušovec [1]

1   Faculty of Civil Engineering, Transportation Engineering and Architecture, University of Maribor, SI-2000 Maribor, Slovenia
2   Carinthian Regional Museum, SI-2380 Slovenj Gradec, Slovenia
*   Correspondence: rok.kamnik@um.si

**Abstract:** Modern heritage protection goes beyond the mere protection of individual buildings and objects. Modern technologies and techniques of field data capture and visual (3D) presentations are increasingly penetrating this field and are becoming more and more essential and necessary for archives, cadastres, and users and visitors of museums, exhibitions, collections, and archaeological parks. In the area between Kotlje and Ravne na Koroškem, Slovenia, in 1476–1477, 9 to 10 anti-Turkish fortifications, called Turške Šance, reportedly were erected. The remains were left to decay slowly. This paper highlights the possibility of applying interdisciplinary data capture and 3D visualization techniques that are used in the fields of civil engineering and architecture for digital reconstruction of the anti-Turkish fortification as a case study in order to present them in the most contemporary way and emphasize them on a local, regional, national, and international level. Unfortunately, similar remains elsewhere in Europe are primarily ignored (with some notable exceptions). The digital reconstruction of anti-Turkish watchtowers therefore represented an extended reconstruction to revive that part of the historical heritage of Slovenia using the proposed techniques.

**Keywords:** anti-Turkish fortification; 3D visualization; watchtower; tschartake; moat; trench; mound; čardak; Çardak; digital archaeology

## 1. Introduction

Archaeological studies include the documentation and investigation of archaeological vestiges and the development of virtual recreations and reconstructions [1].

In this paper, the focus is on the type of virtual reconstruction in which computer graphics are used to reconstruct nonexistent historic objects to provide a better understanding of and generate hypotheses and interpretations of various evidence. Further, we propose an applicable framework with steps for the virtual archaeological reconstruction process of small-scale historic monuments with an interdisciplinary scenario in which accessible architectural and civil engineering tools are used for surveying and 3D modeling.

Virtual reconstruction is an archaeological and architectural field that has transitioned to the digital realm in recent decades [2]. Virtual, which means "potential" and conveys the likelihood of an object having existed in the past, comes from the Latin word "virtus". Such reconstruction predates the invention of the computer and is not just a digital issue. The Envois de Rome of the French Academy of Sciences provides strong support for the theory of reconstruction in archaeology and building [3].

According to El-Hakim et al. [4], there are numerous reasons for the 3D reconstruction of heritage sites, the most important of which are: reconstructing historic monuments that no longer or only partially exist; visualizing scenes from perspectives that are impossible to achieve in the real world; interacting with objects without risk of damage; and providing virtual tourism and exhibits. Therefore, both experts and the general public are already aware of 3D reconstructions of historic structures, even entire towns. A significant body of literature has also been written on the advantages and disadvantages of this method [5].

The first commercial 3D software package, Wavefront Technologies, was introduced in 1984 to meet the expanding demands of motion pictures, after which three-dimensional computer graphics techniques grew in popularity in the television and film industries. The earliest recorded work was that of the bath building at Caerleon Roman Fort in South Wales [6]. A year later, the Old Minster of Winchester's animated virtual tour became the first of its kind [7].

A wide range of cultural organizations, including museums, are now able to apply interactive techniques and information technologies due to the advancement of their software and hardware as well as a reduction in their prices. A lack of exhibition space, high exhibition costs, and the fragility of some artifacts that museum administrators desire to safeguard against potential damage were all addressed by these new technologies. To visualize the cultural background of museum exhibitions, curators have acknowledged and successfully utilized the significance of the new methodologies and instruments [8,9]. Furthermore, museum curators use these new technologies to digitize information on exhibition artifacts and to display and spread cultural information to the public in an appealing and effective manner [8].

According to Demetrescu [2], the reconstruction pipeline shown in Figure 1 begins with the gathering of all the facts about a monument on the field (survey or excavation). All accessible sources are gathered in addition to the work being done on the field, including old sketches, pictures, and data from situations that are extremely similar. The so-called dossier comparatif [10] is a convenient place to store and organize all of these details. The next step is to use the dossier comparatif to produce the eidotipi, sketches, or technical drawings using digital tools [11], during which the researcher can make any necessary corrections to their initial hypothesis before beginning to model in 3D. In this scheme, the 3D model appears to be the final stage and the result of the entire procedure. If there is an "incongruity", as a result, the 3D reconstruction hypothesis must be changed. The simulation serves as a test of the accuracy of the reconstruction; the researcher must make changes to the dossier comparatif or eidotipi or simply conduct more research or find other sources of study.

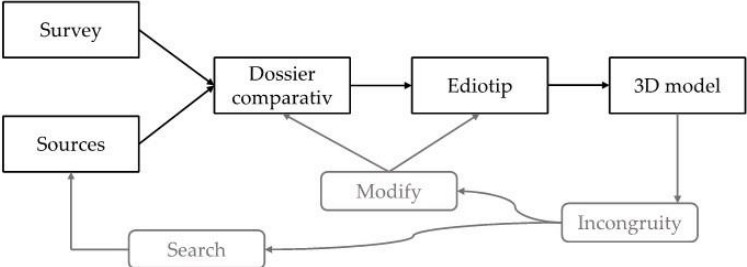

**Figure 1.** Archaeological theory in virtual reconstruction according to Demetrescu [2].

Both reality-based modeling, which is the digital acquisition through a 3D survey of existing archaeological contexts [2], and source-based modeling, which is the virtual restoration of nonexistent archaeological contexts [12], can be used to create 3D content for cultural heritage. In the first instance, the model's "accuracy" is quantitative, whereas in the second instance, the accuracy is qualitative because it is derived from sources with different degrees of reliability.

The goal of this research was to propose the steps involved in heritage visualization, including the sources that were chosen and how they were used in the virtual reconstruction, rather than to suggest solutions for the visualization of the 3D model's degree of reliability.

According to Demetrescu [13], virtual reconstruction is occasionally confused with mesh reconstruction or postprocessing of a digital capture. It consists of a number of phases that include documenting, interpreting, and visualizing missing archaeological contexts. Although the scientific world has acknowledged the promise of this application [13–18], there are not many case studies on reconstruction in the literature, and its contributions

to the incorporation of 3D modeling in archaeological research methods are not very common either. Only 20% of studies on the application of 3D technology in archaeology, according to Münster and Koehler [19], concentrated on the 3D reconstruction of lost contexts. Additionally, most of the initiatives covered in 452 journal papers and conference proceedings included constructing models for already-existing structures and collecting data. Only 16% were concerned with architecture that is no longer in existence, which is intriguing for research on the connections between conventional humanities and digital technologies [19]. Additionally, most authors were connected to institutes that deal with computing (70%) whereas only 14% and 9% respectively come from the engineering and architectural professions. Due to this, virtual reconstruction of lost heritage is still a relatively undefined discipline in the field of archaeological research and its methodology is still highly dispersed in terms of data transparency and acceptance.

However, technical workflows are well established and comparable to other 3D modeling chores such as engineering and design for a VR or CAD modeling of nonexistent objects. Dealing with historical sources or transdisciplinary workflows are more often the specific challenges for these interpretative reconstructions [19].

Therefore, the goal of this paper was to propose an applicable interdisciplinary framework with steps for the virtual archaeological reconstruction process of small-scale historic monuments that no longer or only partially exist.

### 1.1. Digital Archaeology and Interdisciplinary Methods

Today, the practice of making digital replicas of artworks and restoring and recontextualizing them within artificial simulations is widespread in the virtual heritage domain [20]. The modern audience increasingly relies on audio–visual aids to absorb complex ideas or stories quickly [21]. Visual reconstructions of archaeological sites and materials have been around since before the formal construction of archaeology as a discipline itself. However, there has been an expansion of the methods of reconstructing and representing the past in recent decades due to the use of digital technology [22]. The 3D modeling of archaeological sites and artifacts can generate aesthetically pleasing visualizations; nevertheless, considerations of scientific accuracy, ethics, and educational value are needed. From a scientific point of view, it is also important to show the process, appropriate documentation, and used source materials [23].

The use of visual aids and digital media in archaeology is critical not only for public dissemination, but also within the academic community. As a result, museums, cultural institutions, and government agencies should revise their public-interest strategies for history, archaeology, and the environment. Archaeologists are borrowing tools, techniques, and theories from other disciplines to improve the way they collect, analyze, and disseminate archaeological data. Digital media and technology provide a variety of novel and creative methods for capturing public attention and increasing overall competency and appreciation for the past [24,25]. Modern 3D software tools can help with heritage visualization production. They can significantly improve visuals and aesthetics for the presentation of a holistic image of the past, even if they are mostly employed for animation, gaming, and architecture [26].

Therefore, this study highlighted the possibility of applying interdisciplinary data capture and 3D visualization techniques being used in the fields of civil engineering and architecture for digital reconstruction of an anti-Turkish watchtower as a case study, as well as an overview of the practical process of performing such science-based archaeological 3D reconstructions and visualizations, so that they are constructed and presented in the most scientifically sound, informative, and entertaining manner possible in order to ultimately inform and engage the wider public. According to Lopez-Menchero and Grande [27], as long as computer-based visualizations are utilized to enhance archaeological heritage rather than to draw attention away from the actual site or an item in a museum, it is beneficial. Furthermore, if the artifact or location is appropriately introduced and contextualized

with the significance of the legacy to a larger historical discourse, there may be a higher appreciation for the object or location [25].

### 1.2. Case Study—Turške Šance in Slovenia

The system of anti-Turkish trenches (mounds) and fortifications (towers), which is said to have been mentioned already at the end of the 15th century, was used as a case study. The chronicler Jakob Unrest wrote in his *Austrian Chronicle*, which covers the period from 1452–1499, that in 1476 the lords of the land collected a tax which they used to build walls and military outposts to defend against Turkish incursions, starting with a long barrier with outposts near Ravne in Carinthia [28]. Unrest is later quoted by many authors as constantly repeating the following phrase: "Ein lange Lanndt Wer zw Guettenstaynn mit Posteyn" [29], which they translated to mean that at Ravne, there were long barriers with guardhouses [30] that were built by locals after the Turkish invasion in 1476, according to Unrest.

The construction was thought to have taken place at the end of 1476 and the beginning of 1477 [31–33]. There were said to be 9 to 10 fortifications in all. They were placed along the old road from the Ravne manor to the church of St. Mohor and Fortunat in Podgora. The church, which had already been damaged before, was secured with a moat. The Grinfels manor was included in the new anti-Turkish valley barrier [34], which started on the left bank of the Meža river and continued to the foot of the mountain Uršlja gora to the Dvornik farm in a total distance of just over 4 km. In July 1478, the Turk forces returned from Carinthia with loot and many captives, passing Slovenj Gradec [35]. What happened to the Turkish trenches after 1478 is not known.

The first preserved map of the Turkish trenches dates from the second half of the 16th century [36] (see upper right corner in Figure 2a—due to cartographic reduction, only five are depicted); they are also drawn on the Franciscan-Josephine cadastre (third military measurements (1769–1787)) on the Franciscan map, but from 1825, they are not marked. Some similar watchtowers on the Kolpa River between todays Slovenia/Croatia state border can also be seen in Figure 2b by Martin Stier from 1664.

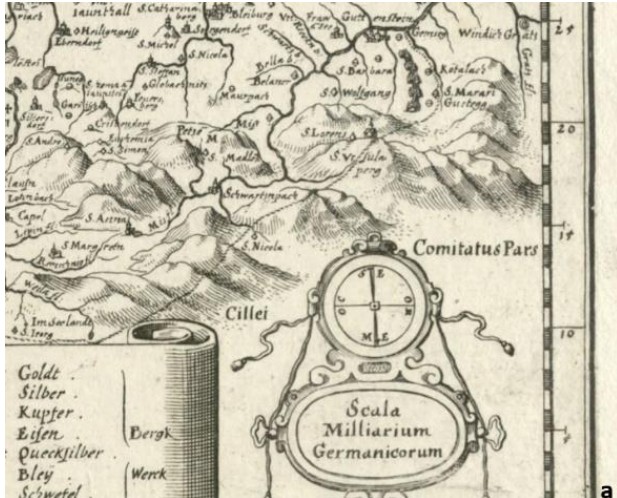 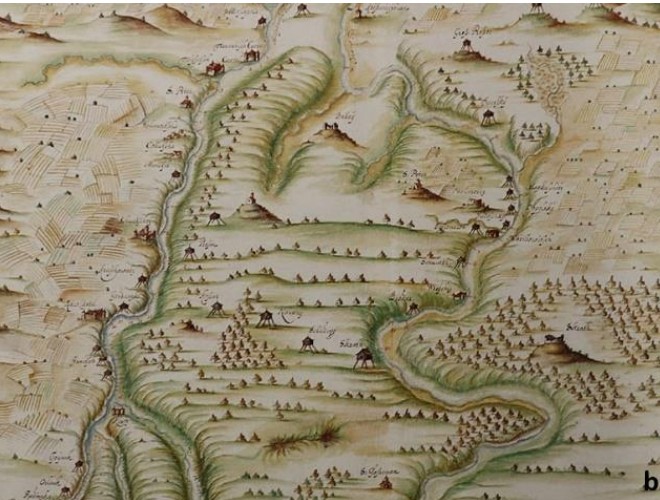

**Figure 2.** (**a**) A Carinthia map in the second half of the 16th century (I. Holzwurm (1575–1617)) [36]; (**b**): watchtowers on the Kolpa river, Slovenia/Croatia state border [37].

Most trenches today have a diameter between 30 and 40 m; the size of the central space, where the watchtowers were supposed to stand, varies between 5 × 5 to 6 × 6 m. The exception is double trench number 6 (TŠ 6), which is larger and where a military crew could be accommodated in the wooden watchtower in the middle plateau. The shape of the wooden watchtower or "Čardak", as we know it from many Croatian sites and the later Vojna Krajina [11–39], seems to be the most likely. It is interesting to see that the same word

"Chardak" is today also used for a balcony with the windows closed [40]. The word derives from a Persian chahartaq (having four arches) (in German, tschartake; in Turkish, Çardak), meaning a watchtower and an important element of the fortification systems in the time of the Ottoman Empire. The term was also known in the mid-east area [41].

In Posavje, the Čardaks stood only on the left, Habsburg side of the bank of the Sava River, while on the right side of the Ottoman Empire stood the so-called caravels [38]. In addition to Croatia, Čardaks were built on the territory of Carniola, Styria, Carinthia, and Hungary [38]. The oldest type of Čardak was square [38] and was first mentioned in Croatia in 1521 and Styria in 1522 [39].

Research on Croatian sites and reconstructions of Čardaks along the Austro-Hungarian border (e.g., Hohenbrugg in the valley of the Raba River and Burgauberg-Neudauberg (Austrian part of Gradiščanski)) would indicate the most probable appearance of Slovenian watchtowers if they were reconstructed. Except for the double larger mound, the area of the central plateau on all the other moats, where watchtowers could have been built, was approximately $4 \times 4$ to $5 \times 5$ m in size (surface area, therefore between 20 and 25 m$^2$). The watchtowers stood on four corner pillars with a diameter between 20 and 25 cm and a height of approximately 3 m [12,39]. A wooden house was built on top of these pillars. The house had a wooden floor with a central opening for lowering and raising a ladder, through which the guards could climb into the upper part. The opening could be closed with a wooden flap if necessary. The floor and walls of the house were built from horizontal planks. On the wooden floor of the guardhouse, four supporting pillars for the roof structure and a protective wooden fence were placed (parapet).

The walls were closed only up to about two-thirds; the rest was open on all four sides with larger rectangular openings for observation of the surroundings. The wall was additionally protected from the inside with narrower vertical boards. The roof was covered with oak shingles [39] (in our area, more likely larch). Oakwood was mainly used for the construction of the Čardak and roof. Boards, posts, and shingles were attached with variously shaped forged iron nails. The lower part of the Čardak was secured with a fence with sharpened stakes attached to the pillars. Part of the fence had to be moved to access the porch. There was room for 6 to 10 guards in such a Čardak. In the house itself, we could expect a wooden bench, modest beds (bags of hay), a wooden chest for storing weapons/earthenware/lamps, a movable ladder and perhaps an even smaller earthen stove for cooking/heating, remains of lead grains for guns, metal parts, military boots, etc. [39].

Horses were tied up near the Čardak, and there had to be a place to light a bonfire with prepared brambles and branches. Signaling could also take place by shooting or ringing bells in churches [39]. Among weapons, Matijaško [39] lists personal cold armaments (e.g., knives), mortars, rifles (matches), and long spears.

The guardhouse was therefore protected first by a high embankment, then by a ditch with stagnant water, further by the steep bank of the central elevation, and then by the elevation of the guardhouse from the ground. In addition, as the terrain's configuration shows, the guardhouse with other trenches in more exposed places was most likely protected in areas with a wooden palisade.

The form of Čardaks was preserved until the 18th century. Most of them were built after the peace agreement signed in Srijemski Karlovci (Serbia) in 1699 between representatives of the Holy League and the Ottoman Empire [39]. Croatia's only reconstructed Čardak (younger, from the 18th century) is in the Lonjsko field Nature Park in the Krapje Dol ornithological reserve.

## 2. Archaeological Context—Similar Watchtowers in Europe

At least 33 similar constructions or remains were found across Europe (Table 1). Most of them in are in Croatia (15), Serbia (7), Germany (5), Austria (3), and BIH (2); 1 is unknown. Most of them, according to gathered data, were built in the 16th or 17th century.

Their positions are also visible in Figure 3. Some location data and construction years were unavailable.

**Table 1.** Čardaks across Europe.

| No. | Place | Country | E | N | Year or Century |
|---|---|---|---|---|---|
| 1 | Lütjenburg [42] | Germany | 10°34′18.4″ | 54°18′05.1″ | / |
| 2 | Dragići [43] | Croatia | 17°18′44.9″ | 45°12′24.1″ | / |
| 3 | Senj [44] | Croatia | 14°54′32.5″ | 44°59′23.0″ | 16th cent. |
| 4 | Otočac [44] | Croatia | 15°13′47.4″ | 44°52′07.8″ | 16th cent. |
| 5 | Slunj [44] | Croatia | 15°34′55.0″ | 45°06′57.3″ | 16th cent. |
| 6 | Glina [44] | Croatia | 16°05′32.8″ | 45°20′25.3″ | 16th cent. |
| 7 | Hrastovica [44] | Croatia | 15°08′47.8″ | 45°58′08.8″ | 16th cent. |
| 8 | Sisak [44] | Croatia | 16°22′58.2″ | 45°28′34.0″ | 16th cent. |
| 9 | Ivanić [44] | Croatia | 16°23′36.9″ | 45°42′32.4″ | 16th cent. |
| 10 | Koprivnica [44] | Croatia | 16°49′25.4″ | 46°10′03.7″ | 16th cent. |
| 11 | Križevci [44] | Croatia | 16°32′18.5″ | 46°01′58.2″ | 16th cent. |
| 12 | Đurđevac [44] | Croatia | 17°03′48.5″ | 46°02′27.6″ | 16th cent. |
| 13 | Drnje [44] | Croatia | 16°55′55.2″ | 46°12′48.7″ | 16th cent. |
| 14 | Stupanj [45] | BIH | 19°05′25.7″ | 44°48′20.2″ | / |
| 15 | Subotište [45] | Serbia | 19°57′24.5″ | 44°50′53.0″ | / |
| 16 | Ada [45] | Serbia | 20°06′47.9″ | 45°48′57.7″ | / |
| 17 | Lisačka [45] | Serbia | / | / | / |
| 18 | Majur [45] | Serbia | 19°38′56.8″ | 44°46′07.6″ | / |
| 19 | Bosut [45] | Serbia | 19°21′47.0″ | 44°56′37.2″ | / |
| 20 | Protina Bašta [45] | / | / | / | / |
| 21 | Beli Breg [45] | Serbia | 21°49′09.1″ | 43°28′30.1″ | / |
| 22 | Petrova gora to | Croatia to | 16°01′36.6″ | 46°09′53.9″ | 1669 |
| 23 | Novi Grad [46] | BIH | 16°23′28.8″ | 45°02′14.3″ | 1669 |
| 24 | Stara Gradiška to | Croatia to | 17°14′37.9″ | 45°08′55.9″ | / |
| 25 | Zemun [46] | Serbia | 20°17′28.7″ | 44°52′52.6″ | / |
| 26 | Pforzheim [47] | Germany | 8°46′55.9″ | 48°54′16.0″ | 1695–1697 |
| 27 | Ötisheim [48] | Germany | 8°49′53.7″ | 48°58′07.4″ | 1695–1697 |
| 28 | Sulzfeld [48] | Germany | 8°52′03.6″ | 49°05′05.0″ | 1695–1697 |
| 29 | Eppingen [47] | Germany | 8°56′25.5″ | 49°06′40.0″ | 1695–1697 |
| 30 | Lafnitz [49] | Austria | 16°01′26.3″ | 47°22′27.3″ | 1700 |
| 31 | Burgau [50] | Austria | 16°04′53.8″ | 47°08′30.4″ | / |
| 32 | Fehring [51] | Austria | 16°00′49.8″ | 46°56′12.3″ | 1706 |
| 33 | Vikići [52] | Croatia | / | / | / |

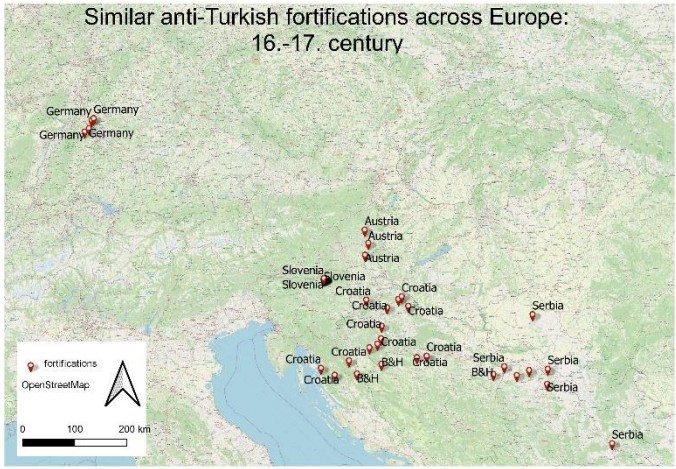

**Figure 3.** Similar watchtowers across Europe.

Some reconstructions have been made. Figure 4 shows examples from several places in Germany, Austria, Serbia, and Croatia. According to the literature, the Croatian rebuilding is the most similar to the fortifications that could be erected in Slovenia previously. The watchtowers were recreated using materials that were most likely used at the time of their creation (larch or oak). However, reconstructions of buildings that have been demolished are rarely carried out. In such cases, cheaper methods such as 3D modeling can be used. A good example is the creation of a 3D model of the altars and interiors of the Çatalhöyük houses in Turkey [53], the church of San Nicolò, Italy [54], or the recreation of the lararium of the Roman domus of Torreparedones [55].

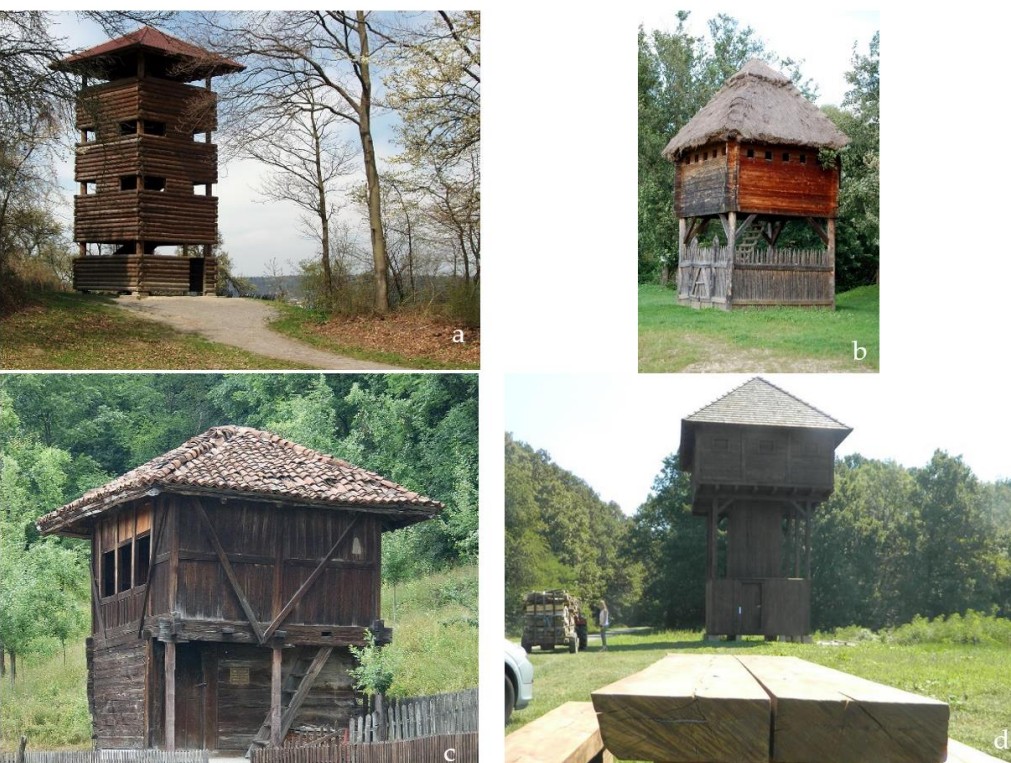

**Figure 4.** Some 1:1 reconstructions: (**a**) Niefern-Oschelbron, Germany; (**b**) Burgau/Lafnitz, Austria; (**c**) Šumadija, Serbia; (**d**) Dragalić, Croatia.

## 3. Civil Engineering Context

### 3.1. Fieldwork Methods

3.1.1. Aero-Photogrammetry Modeling

With the help of the point cloud obtained using a DJI Mavic Pro drone (Shenzhen DJI Sciences and Technologies Ltd., Shenzhen, PRC), we created 3D models of the terrain using different programs (Pix4D (S.A., Prilly, Switzerland), Autodesk Recap EDU ver. 6.2, and Recap Photo EDU ver. 20.3.1.47 (Autodesk Inc., San Francisco, CA, USA)). This photogrammetry software uses images to generate point clouds, digital surface and terrain models, orthomosaics, textured models, and more. A digital elevation model, which is a model that contains all elevations such as trees, roofs of buildings, etc., was also created.

With the help of the drone photos, a 3D model of the moat was created in Recap Photo. Official free Lidar recordings were also available to us. With the use of the Recap program, a 3D model of the surroundings also was created (Figure 5).

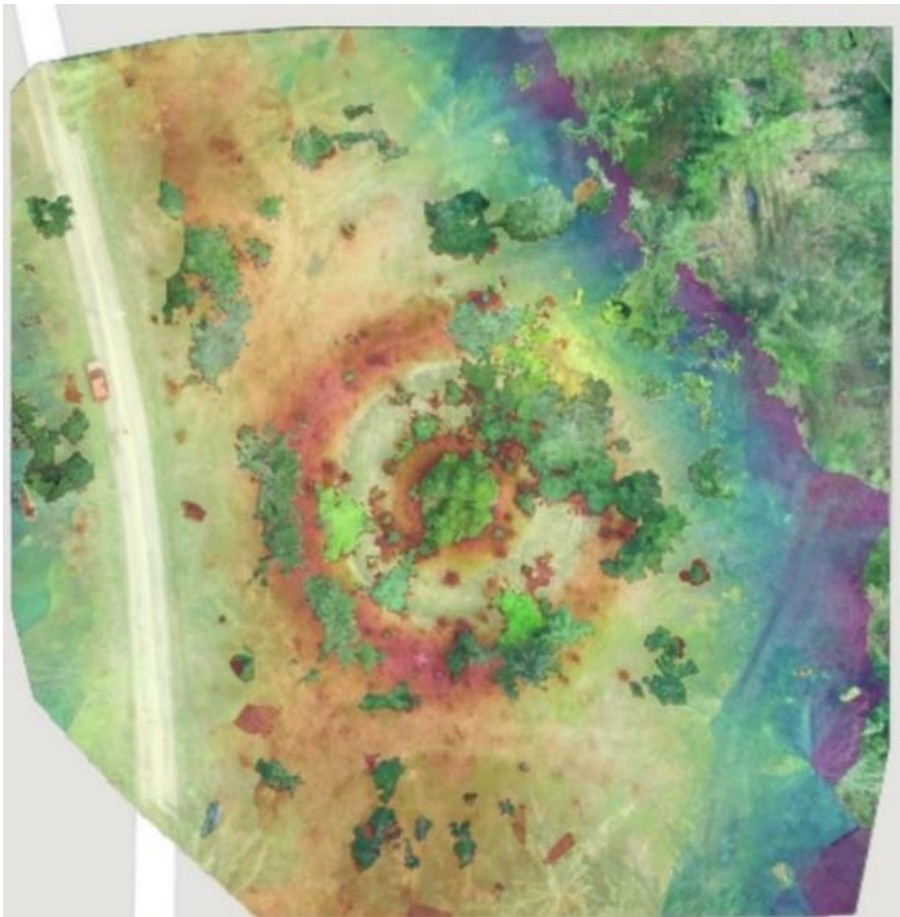

**Figure 5.** 3D model of TŠ3 trench.

3.1.2. Archaeological Excavations

The excavation was carried out at the location of the third moat (TŠ3) in 2020. The archaeological research aimed to determine what materials the watchtower was made of, how it was built, and the age of the objects found. Two probes were opened (see the geodetic plan in Figure 6). The work in Probe 1, which ran over the outer embankment, was carried out mechanically. On the central plateau, excavations in Probe 2 (Figure 7) were carried out exclusively by hand due to the steep bank of the trench and standing water in the ditch. The depth of standing water in the ditch was between 0.3 and 0.5 m at that time. Probe 2 was placed along the entire length of the plateau (it occupied more than 38% of the surface) in the least forested part (see Figures 6 and 7). The double ditch (TŠ6), given its position at the top of the ridge and the double ditch that surrounded it, was at least intended for a permanent or occasionally inhabited military crew that had to stay in the watchtower. Unfortunately, TŠ6 could not be explored further at the time because it was not fully accessible. In Probe 1, no traces of fortifications or the remains of some stakes that would additionally protect the tower were found. The oldest and only discovery during the research was a late medieval clay roof tile from Probe 1. The absence of archaeological records (traces) may indicate that a guardhouse was not built on this trench (TŠ3) but may also mean that the watchtower was not fully built.

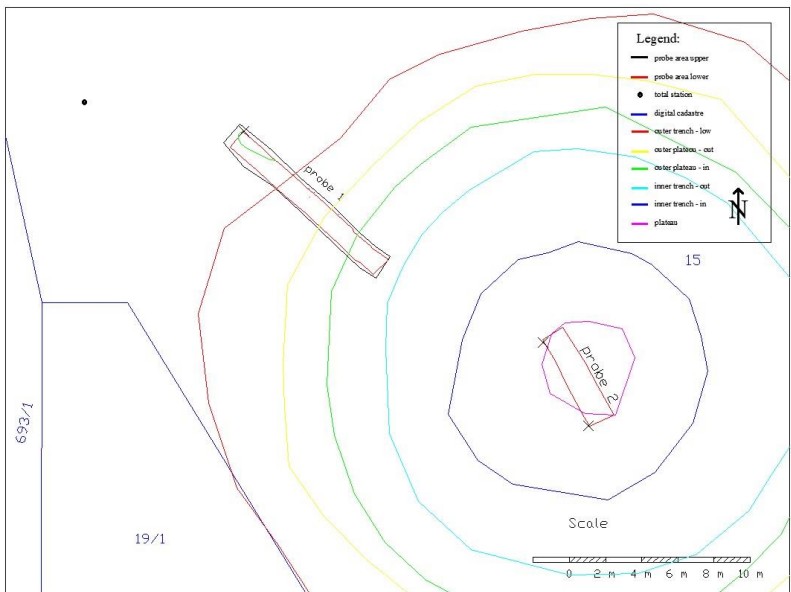

**Figure 6.** Geodetic plan of excavations.

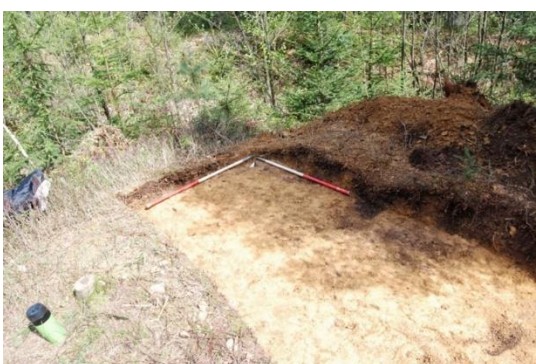

**Figure 7.** Probe 2 (view to the north).

### 3.1.3. Metal Detector Investigation

The entire area of TŠ3 was also investigated using a metal detector. Seventeen points of potential interest were identified. All were located on the embankment; the sensor did not detect any metal remains on the central plateau. After surveying all the points, the objects were excavated. Most were scraps of various aluminum foil, cans, and bottle caps. A post-war copper hunting cartridge (RWS 7 × 64), a Yugoslavian five para coin (item T216), and an iron nut from a tractor (item T215) were found. All finds were between 2 and 10 cm deep in the humus layer of Probe 1.

### 3.2. Static, Material, and Cost Calculations

In the territory of the central and southern Balkans, there has been a square tower form of architecture built of stone (kula or tower house) since the 13th century. They served both civilian (residential buildings) and military purposes. The phenomenon of the extended family typical of Southeastern Europe, in which the home was often protected, gave rise to the kula or tower house [56].

Čardak, as already explained in the introduction, is the Turkish word for a wooden building on four pillars. At the time of the Ottoman invasions, they were the most widespread form of wooden guardhouses in the wider territory of Serbia, Croatia, Austria, and Germany. They were intended for reconnaissance, so their shape was most probably based on the kula or tower house. They were used to monitor the movements of Turkish troops and alert the local population.

Wooden construction was typical for this period in our area, both in the countryside and in cities (the exception was the castles or mansions of the upper classes) [57]. Building with wood was cheaper and the consumption of wood for a Čardak was small. Wood began to run out in our country in the 17th century (due to glassworks and ironworks). Wood was also more accessible (it could be cut and processed in the immediate vicinity), there was almost no transport, and the technological process was simple.

The structural safety assessment of heritage objects is a common process in assessing the condition of the structure and is needed in the cases of reconstruction, renovation, and/or rebuilding. The cases such as Torre de la Vela in la Alhambra, Granada, Spain [58]; Qutb Minar, India, as one of the tallest stone masonry towers [59]; the medieval masonry bell tower in the Cathedral of Fiesole, Italy [60]; and churches after the earthquake [61] are good examples of including static analysis in the heritage building research.

Regarding the history of construction, the characterization of the construction materials, seismic assessment, and static and dynamic monitoring, many studies have been carried out in the Mallorca Cathedral. They included historical investigations of the building's development, examination of the soil beneath it and its structural components, structural assessments using both straightforward and sophisticated methods, and monitoring [62]. Furthermore, a study by Gençer [63] aimed to identify factors influencing structural resistance and failure mechanisms of ashlar Cilician dry masonry watchtowers under lateral stress. Then, by using the quasistatic tilting approach, virtual towers were created based on the characteristics found in the case study.

Another study by Elyamani in 2018 [64] aimed to provide a proposal for the reuse of the Baron Empain Palace in Cairo. To support this reuse proposal, a 3D numerical model of the palace was created and the new expected loads were applied on it. It was discovered that the palace's walls and foundations could withstand the new loads. The slabs were discovered to be unable to sustain the new loads in some places; further investigation and analysis are required to determine their actual capacity.

For this study, the static calculation of the digitally designed watchtower was carried out according to Eurocode regulations and the analysis was carried out using the SCIA Engineer program. The model can be seen in Figure 8. All possible loadings were considered: constant load, payload, wind load, snow load, and earthquake load. The roofing, pillars, rod arms between the platform and the column, and the platform were also dimensioned. A material utilization review was also conducted.

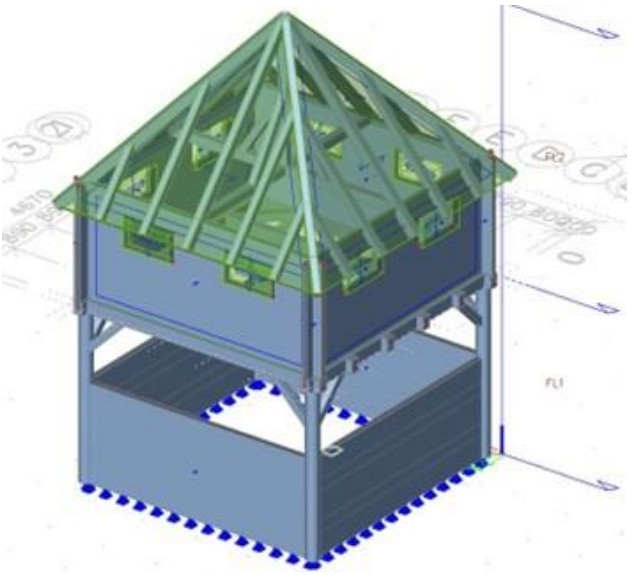

**Figure 8.** Static calculations for a watchtower.

Each element that made up the watchtower was listed as a segment of the wooden structure along with its dimensions and the number of individual pieces; only a part was given in m² for a more straightforward interpretation and cost estimation (Table 2).

**Table 2.** Elements of the watchtower.

| Element | Dimensions (cm) | | | Number (Pieces) |
|---|---|---|---|---|
| | Width | Height | Length | |
| Lower part | | | | |
| Support column | Ø = 26 cm | - | 300 | 4 |
| Supporting diagonal | 12 | 12 | 80 | 8 |
| Cross board (bottom) | 3 | 20 | 427 | 8 |
| Pillar fence | Ø = 8 cm | - | 170 | 97 |
| A stake with a point | 8 | 8 | 210 | 101 |
| Pointed pillar (door) | Ø = 16 cm | - | 230 | 2 |
| Diagonal (door) | 10 | 1.5 | 110 | 2 |
| Upper part | | | | |
| Board—wall | 8 | 15 | 490 | 52 |
| Board—floor | 20 | 5 | 434 | 22 |
| Crossbars (platform) | 16 | 22 | 450 | 8 |
| Stick hands | 12 | 12 | 132 | 4 |
| Roof (inclination 45°) | | | | |
| Rafter | 12 | 16 | 333 | 8 |
| Beams | 12 | 16 | 450 | 6 |
| Stick hands | 12 | 12 | 132 | 4 |
| Roofing (shingles) | area: | | 64.45 m² | |
| Substructure for shingles | 4 | 5 | 305 | * |
| Ladder | | | | |
| Pillar | 6 | 15 | 410 | 2 |
| Walking crossbars | 10 | 6 | 75 | 8 |
| Shutters | | | | |
| Horizontal slats (frame) | 15 | 4 | 100 | 16 |
| Vertical slats (frame) | 10 | 3 | 42 | 16 |
| Cross slats | 9.5 | 3 | 80 | 32 |
| Diagonal | 10 | 1.5 | 41 | 8 |
| Bridge | | | | |
| Boards | 20 | 5 | 200 | 92 |
| Boards (construction) | 8 | 16 | 500 | 6 |
| Transverse beams | 8 | 16 | 200 | 8 |
| Pillars | Ø = 25 cm | - | 550 | 5 |
| Pillars | Ø = 25 cm | - | 190 | 4 |
| Slats (fence) | 10 | 15 | 478 | 6 |
| Slats (diagonal) | 10 | 10 | 85 | 24 |
| Substructure for walking boards | 8 | 16 | 500 | 5 |
| Gravel fill | | 0.59 m³ | | |
| Foundation | | | | |
| Point foundations | 80 | 80 | 80 | 4 |

* The number of slats (dimensions 4 × 5 × 305 cm) that served as a substructure for the selected roofing depended on the roof construction itself and the selected shingle dimensions.

The watchtower was thus divided into three segments: the lower part, the upper part, and the roof, which covered the entire tower. A ladder used for vertical communication in the watchtower and shutters were also included on the list. A special section was also dedicated to the bridge and the foundations, where only the approximate values of the individual point foundations were listed.

Sustainable tourism should embrace concerns for environmental protection; social equity; the quality of life; cultural diversity; and a dynamic, viable economy delivering jobs and prosperity for all [65]. Nowadays, when referring to cultural heritage objects,

one of the first aspects implies not only the object itself, but also creating 3D models using different technologies [66]. Nowadays, many researchers explore different methods for documentation, management, and sustainability of cultural heritage, which has become an interdisciplinary approach to the development of culture [67]. A 3D model of cultural heritage is one of the possibilities for sustainable tourism and cultural heritage. In the Strategic Baselines of the Development Cohesion Region of Eastern Slovenia [68] and the strategy of the Regional Development Program for the Carinthia Development Region 2021–2027 [69], one of the main goals in the field of sustainable tourism is the goal of developing and upgrading the basic tourist infrastructure, including the revitalization of cultural heritage buildings. By researching the Turške Šance watchtowers, including their appearance and a detailed analysis of the construction costs, some potential investors should be encouraged to engage in a physical reconstruction.

Table 3 shows the inventory of the needed material necessary for constructing the entire wooden structure, consisting of the previously listed materials.

**Table 3.** Prices for needed materials.

| Purchase Goods | Purchase Price (EUR per Piece) | Quantity | Price (EUR) |
|---|---|---|---|
| Oak support round column Ø = 26 cm, 300 cm | 25 | 4 | 100 |
| Oak round column Ø = 25 cm, 550 cm | 43.85 | 5 | 219.25 |
| Oak round column Ø = 25 cm, 200 cm | 15.95 | 4 | 63.8 |
| Beam (larch) 12 × 12 × 400 cm | 17.6 | 6 | 105.6 |
| Oak board 3 × 20 × 430 cm | 15 | 8 | 120 |
| Larch round pillar Ø = 16, 250 cm | 12.75 | 2 | 25.5 |
| Wood pointed pillar round Ø = 8, 200 cm | 7.47 | 100 | 747 |
| Rectangular stake (larch) 8 × 8 × 400 cm | 8.49 | 101 | 857.49 |
| Oak board 8 × 15 × 490 cm | 33 | 52 | 1716.00 |
| Solid flat larch slats 10 × 1,5 × 200 cm | 6.5 | 4 | 26 |
| Wood—Siberian larch 5 × 20 × 450 cm | 41 | 22 | 902 |
| Wood—Siberian larch 16 × 22 × 450 cm | 48.4 | 8 | 387.2 |
| Beam (larch) 12 × 16 × 400 cm | 23.47 | 8 | 187.76 |
| Beam (larch) 12 × 16 × 450 cm | 26.4 | 6 | 158.4 |
| Beam (larch) 6 × 15 × 410 cm | 14.27 | 2 | 28.54 |
| Beam (larch) 10 × 6 × 100 cm | 5.63 | 8 | 45.04 |
| A flat board made of Siberian larch 15 × 4 × 400 cm | 14.25 | 4 | 57 |
| Wood—Siberian larch 3 × 10 400 cm | 12.21 | 2 | 24.42 |
| Wood—Siberian larch 3 × 9.5 × 400 cm | 11.96 | 7 | 83.72 |
| Wood—Siberian larch 5 × 20 × 400 cm | 36.7 | 46 | 1688.2 |
| Siberian larch tree 8 × 16 × 500 cm | 19.55 | 11 | 215.05 |
| Siberian larch tree 8 × 16 × 400 cm | 15.65 | 4 | 62.6 |
| Beam (larch) 10 × 15 × 500 cm | 23.79 | 6 | 142.74 |
| Beam (larch) 10 × 10 × 400 cm | 12.07 | 6 | 72.42 |
| Roof ** | EUR 72.00/m$^2$ | 64.45 m$^2$ | 4640.59 |
| Natural gravel 0/63 mm | EUR 8.92/m$^3$ | 0.59 | 5.26 |
| SUM | | | 12,681.58 |

** The production of the entire roof structure using the materials found in the list of works and the price per m$^2$ for the entire roof was considered under the assumption that the roof consisted of rafters and horizontal layers and was laid in two layers with nailed shingles. The shingles, as well as the entire roof, were made of Siberian larch wood.

The prices of individual pieces were valid for the period of spring 2020 in the Slovenian territory. They were obtained from technical stores with building materials and other intermediaries of wood products or semifinished products. The prices of semifinished wood products may vary depending on the price changes in the market for forest wood assortments. They also differed in the cases of volume discounts from the technical wood broker or other contractual factors when purchasing semifinished products.

Since there not all the dry wooden semifinished products in the exact dimensions needed for the project are available, the processing of the purchased materials will result in some wasted (unnecessary) parts, which should be managed in an appropriately (ecologically indisputable) way.

In addition to each product's value, the purchase price per piece included the trade margin and the tax, which was 22% according to the Slovenian VAT legislation at the general rate.

## 4. Architectural Context—Results

After collecting the data, a 3D digital reconstruction was prepared. When architects design projects, they must produce a representation for the client that translates their concepts and the structure's requirements. Therefore, the process of creating a visualization is done in terms of tools and materials modified from digital 3D modeling approaches used in architecture. According to Schoueri and Ferreira [22], realistic foundations should underlie the building and its surroundings. The archaeological structures are frequently complex and made up of both old and new constructions in varying stages of development. Therefore, it is important to take care and consider how the visualizations are created and presented.

The modeling process was started using Graphisoft Archicad 23 software (Educational version, Budapest, Hungary). The modeling began with simple blocks that delineated the structure's dimensions as well as known wall heights and the roofing situation. The wall thicknesses and door and window openings were estimated for the structure and were included at this stage.

To create the best possible reconstruction, contemporary analogies to the most common trend for the region and time period were researched. The towers on the Turkish moats were the same as Čardaks in terms of the construction method and the use of materials, and thus served as an example.

The watchtower (Figure 9) was placed on four round pillars made of oak wood on which stood a simple wooden guard room with a square floor plan of 4.5 m × 4.5 m intended for about eight guards. Support columns with a diameter of about 25 cm were buried in the ground. The strengthening foundations for the columns were represented by larger stones, which filled the holes and served for drainage purposes and prevented the rotting of the buried wood. The height of the walls of the guard room corresponded to the average size of a standing man (about 1.9 m). It was made of wooden oak layers, which were used for walls and floors. They were connected in the corner with a carpenter's bond and fastened with forged nails.

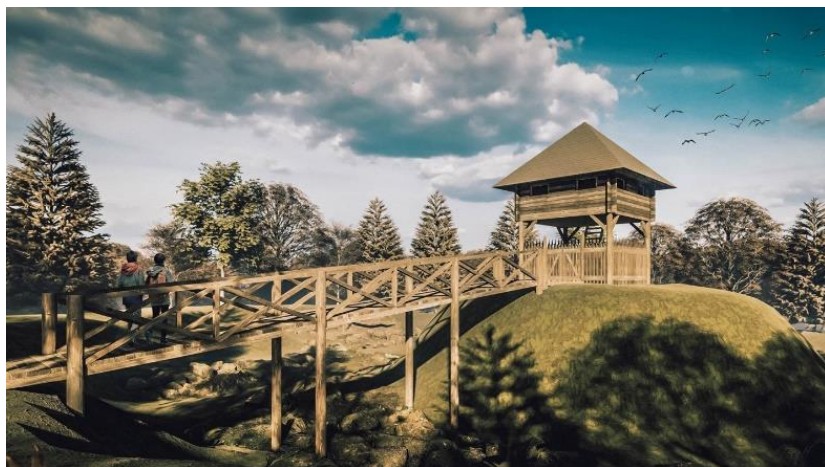

**Figure 9.** The 3D visualization of the watchtower [70].

The upper part of the guard room was built of horizontal wooden beams up to approx. two-thirds. To be able to guard and observe the surroundings, square openings were cut above them in the upper part, which could be closed with simple wooden covers in case of bad weather. On the inside of the house, narrower vertical boards were nailed to these two-thirds. The roof of the guard room rested on four vertical pillars and was covered with oak shingles.

A wooden bridge made of the same materials as the watchtower itself led to the watchtower (Figure 10). Two intermediate supports were added for static stability and a fence for safety.

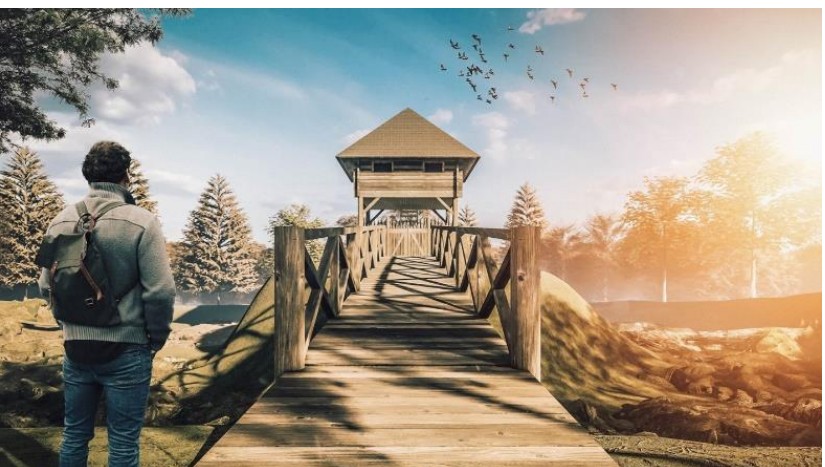

**Figure 10.** Bridge leading to the watchtower over the trench [70].

The interior of the guard room was complemented by the most necessary furniture, namely a wooden bench, a bunk bed, a wooden chest, and a small clay oven for cooking food and pottery (Figures 11 and 12). The selected archaeological objects represented typical pieces of interior design, tools, and weapons from this period [39].

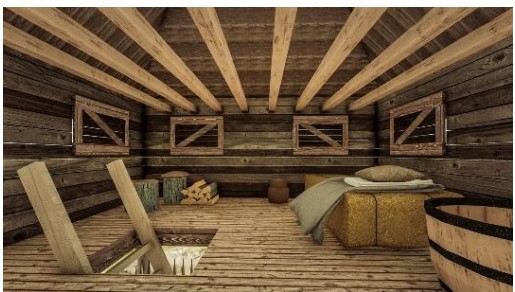

**Figure 11.** Interior of the guard room [70].

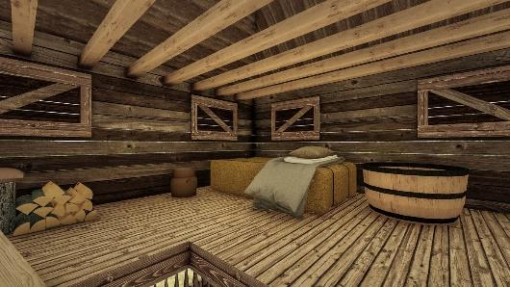

**Figure 12.** Interior of the guard room [70].

## 5. Discussion

As presented in the Introduction, some interdisciplinary reconstructions have been documented in the past decades [19,71] and also in recent years; however, most of them were of existing objects [4,13,64,72,73] The virtual reconstructions that were carried out for objects that no longer or only partly existed [5,74] used complicated frameworks and tools such as the specialized design software Rhinocerous 3D, visual blueprint programming, Agisoft Metashape, or the Stencil Kaarta instrument, or expensive equipment such as a terrestrial laser scanner. Therefore, this study presented a possible interdisciplinary scenario in which the accessible architectural software Archicad ver 23 was used for modeling and Lumion software ver. 10.5 (Leiden, The Netherlands) for architecture was used for visualization along with the use of most common modeling techniques in architectural practice. Furthermore, for the photogrammetry analysis of the land, a DJI drone and official free Lidar data were used for the 3D analysis of the location. Furthermore, manual on-site archaeological excavation and soil analysis with a metal detector were carried out.

Moreover, the reconstruction pipeline [2] was extended with applicable tools for each of the phases, adding to the archaeological theory in the virtual reconstruction (Figure 13). In addition, this paper also focused on the application of tools that are economically very feasible. Such application can be achieved with the tools of any midsized design company.

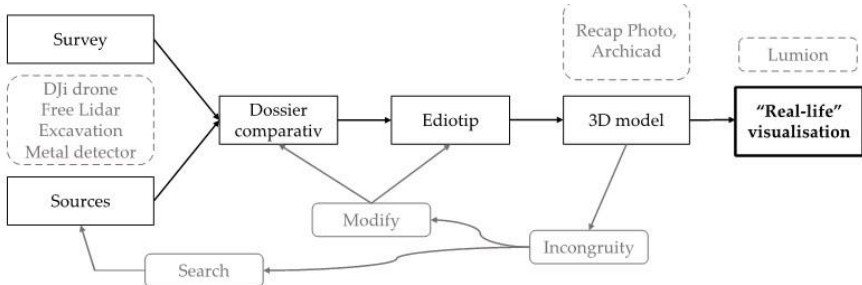

**Figure 13.** Extended pipeline with applicable tools.

## 6. Conclusions

A unique system of anti-Turkish fortifications—defensive ditches that were prepared for the construction of watchtowers, or Čardaks—that has been preserved in Preški Vrh was used as a case study. It was built in the last quarter of the 15th century by the Carinthian provincial estates as a valley barrier against Turkish invasions. The moats were of various shapes and sizes; 10 are mentioned in the literature and 9 are recognized in the field today. Among them, only five have been preserved to their original extent. For the defense of Carinthia, valley barriers were also built in Železna Kapla, in Vrata in Gortina, in Fala in the Drava Valley, on the Jezersko Pass, and on the Ljubelj Pass.

The Turkish moats represent an exceptional cultural and natural heritage (wetlands in ditches). The forest is increasingly overgrowing them and a lot of damage has been done by frost; the moats are being destroyed by cutting down the forest and removing stumps (forest hauling).

Research into the Turkish trenches contributed to a greater understanding of the period in which the Slovenian territory experienced one of its greatest devastations. At the same time, due to the unresponsiveness of the authorities at the time, the farmers had to organize themselves and build a defense system of trenches and watchtowers to protect their property and their lives.

The research revealed that several similar watchtowers were built in Serbia, Bosnia and Hercegovina, Croatia, Austria, and Germany. Some of them were physically reconstructed.

For our virtual reconstruction, an extended reconstruction pipeline was used. A 3D model of one of the remains was made using a DJI Mavic Pro drone and Pix4D, Autodesk Recap, and Recap Phot software.

Archaeologic excavations were also conducted in two probes in the TŠ3 trench. Unfortunately, no physical evidence was found concerning watchtowers or military equipment that would indicate the presence of an army from this period. No particular discoveries were made with the metal detector either.

In addition, a 3D model of a watchtower was created. All needed static calculations were made in the case of the physical reconstruction. All required materials were listed with the exact dimensions and number of pieces. A list of costs was also created. Finally, a 3D digital reconstruction/rendering was created for the watchtower and its interior.

One of the main goals in the field of sustainable tourism is the revitalization of cultural heritage buildings. By researching the watchtowers and their appearances, and by conducting a detailed analysis of the construction costs, some potential investors should be encouraged to engage in a physical reconstruction.

**Author Contributions:** Conceptualization, R.K.; methodology, R.K., S.D.J., M.N.P. and M.J.; validation, R.K., S.D.J. and M.J.; formal analysis, R.K., S.D.J., M.N.P. and M.J.; investigation, R.K., S.D.J., M.N.P. and M.J.; writing—original draft preparation, R.K.; writing—review and editing, R.K., S.D.J., M.N.P. and M.J.; visualization, R.K. and M.J.; supervision, R.K.; project administration, R.K.; funding acquisition, R.K. All authors have read and agreed to the published version of the manuscript.

**Funding:** This research was funded by the European Union; European Social Fund; Investing in Your Future; the Republic of Slovenia; the Ministry of Education, Science and Sport; Student Innovative Projects for the Benefit of Society; Archaeological Park Turške Šance, 2019/20 (11083-6-2019), https://www.srips-rs.si/storage/app/media/RAZVOJ%20KADROV/SIPK/2020%20-%20PROJEKTI/MB/gpa-projekt-1-1.pdf (accessed on 18 July 2022).

**Data Availability Statement:** Data supporting the reported results can be found at: https://www.srips-rs.si/storage/app/media/RAZVOJ%20KADROV/SIPK/2020%20-%20PROJEKTI/MB/gpa-projekt-1-1.pdf (accessed on 18 July 2022).

**Acknowledgments:** The authors are grateful for the valuable contributions and help from the Carinthian Regional Museum, Museum Slovenj Gradec, and Municipality of Ravne na Koroškem.

**Conflicts of Interest:** The authors declare no conflict of interest.

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
