# Peer review of "Using Interdisciplinary Techniques for Digital Reconstruction of Anti-Turkish Fortification Watchtower"

_land, doi:10.3390/land11101756_

Round 1

Reviewer 1 Report (Previous Reviewer 2)

Review of land-1931640-peer-review-v1

The manuscript is a resubmission of an earlier version.

While the authors have made considerable changes and have substantially improved the text, the manuscript still not address the main criticism that I levelled with regards to the first submission. In that review I wrote:

The paper is a straightforward, nice little case study that describes investigations and 3D virtual reconstruction of a guard tower site. While very limited in scope,  the observations are worth putting on record. The paper does not, however, provide any real contribution to theory or application as it is not grounded at all in any literature on reconstructions of archaeological field monuments or modelling. It does not make any serious contribution to the research area of landscape archaeology/history. As it stands, therefore, the paper is wholly unsuited for a journal MDPI Land and must be rejected. The authors should be encouraged to revise the paper and submit it to a more suitable heritage or archaeology journal. I would recommend MDPI Heritage or Archaeologisches Korrespondenzblatt.

For the paper to stand in Land, the authors must provide a detailed discussion of the literature on reconstructions of archaeological field monuments, and the literature on modelling of such  reconstructions. Then they can present their case study. The discussion section then needs to tie the observations back to the literature reviewed in the introduction with the authors highlighting how their work expands on the state of knowledge.

The paper requires a substantial rewrite which can be achieved if the authors commit to it. I hope the authors will do so.

If the authors do not wish to carry out the recommended substantial rewrite, then the paper must be rejected.  In such a case I would then encourage the authors to submit it to a more suitable heritage or archaeology journal. I would recommend MDPI Heritage or Archaeologisches Korrespondenzblatt.

Author Response

Reviewer 2 Report (Previous Reviewer 1)

A. Elyamani, “Re-Use Proposals and Structural Analysis of Historical Palaces in Egypt: the Case of Baron Empain Palace in Cairo,” Ahmed Elyaman

Also in No 48, 50 take out the names beside the title of journals

Round 2

Reviewer 1 Report (Previous Reviewer 2)

The authors have addressed my concerns

This manuscript is a resubmission of an earlier submission. The following is a list of the peer review reports and author responses from that submission.

Round 1

Reviewer 1 Report

11)      Line 14:”in the last third of the 15th century….”: Better use exact age range…1470-1500?
22)      Line16: “..the importance of the anti-Turkish fortifications present them in the most 16 modern way,..”: change wording..importance, modern way… or something else?A fortification is fortification to avoid attacks. The masonry of other architectural element is different. Be specific
33)      L.20 “..The Turkish 20 trenches, therefore, represent an excellent opportunity to revive the archaeological heritage of Slovenia and Europe.” : This is an exaggeration that Slovenia’s and Europe’s valuable heritage depends on Turkish/Ottoman occupation reminiscent! To the contrary it reminds an imprerialistic militaristic approach without any trace of culture that deprived Southern Balkans and up to a degree SE Europe from development. Thus, this part needs rephrasing and do not attempt to present a castle with more value than it deserves.
44)       In the introduction too much descriptions and terms on fortification , Chardak, trench, etc that the reader is confused or must be a local to understand those In my opinion a figure of trenches and watch towers must be inserted here.
55)      No mention what is the basic material for these towers.
66)      Watch towers are existing in much earlier years in history of the Balkan area. Made by stone. A mention of those preserved towers must be connected to those here.
77)      Fig 2 what is the material used?
88)      Regarding the history of construction, the characterization of the construction materials, the seismic assessment, the static and the dynamic monitoring, with 3D references are missing: I suggest Ahmed Elyamani and Pere Roca (2018) SCIENTIFIC CULTURE, Vol. 4, No. 2, 1-24 DOI: 10.5281/zenodo.1214557, one century of studies for the preservation of one of the largest cathedrals worldwide: a review.
9     Elyamani 2018 re-use proposals and structural analysis of historical palaces in egypt: the case of baron empain palace in cairo , in SCIENTIFIC CULTURE, Vol. 4, No 1, 53-73. DOI: 10.5281/zenodo.1048245
FFunda Gençer, 2019, structural characteristics of ashlar roman watchtowers in cilicia region, Anatolia, 10)                Mediterranean Archaeology and Archaeometry Vol. 19, No 3, 63-78 DOI: 10.5281/zenodo.3541100

1Mohammed Al-Nasarat and Abd alrzaq Al-Maani, 2014 PETRA DURING THE CRUSADER PERIOD FROM THE EVIDENCE OF AL-WUAYRA CASTLE: A REVIEW. Mediterranean Archaeology and Archaeometry, Vol. 14, No 1, pp. 221-234.

Reviewer 2 Report

land-1859847-peer-review-v1

Digital reconstruction of Anti-Turkish fortification “Turške Šance” watch tower in Slovenia

The paper is a straightforward, nice little case study that describes investigations and 3D virtual reconstruction of a guard tower site. While very limited in scope,  the observations are worth putting on record. The paper does not, however, provide any real contribution to theory or application as it is not grounded at all in any literature on reconstructions of archaeological field monuments or modelling. It does not make any serious contribution to the research area of landscape archaeology/history. As it stands, therefore, the paper is wholly unsuited for a journal MDPI Land and must be rejected. The authors should be encouraged to revise the paper and submit it to a more suitable heritage or archaeology journal. I would recommend MDPI Heritage or Archaeologisches Korrespondenzblatt.

I am adding some comments below that may help the authors when revising the paper

Line 30 ‘prisons’  I am not sure that this is correct, It seems a translation error by the authors

Line 33 The text “Ein lange Lanndt Wer zw Guettenstaynn mit Posteyn” refers to a moat barrier at that were first described in 1881, this paper MUST be drawn on and cited:

Reiner, J. (1881). Wallschanzen bei Guttenstein in Kärnten. Carinthia I, 71, 50-57

Line46 It would be good to reproduce a section of the map

Figure 4 cannot be read and needs to be bigger

Line145-146 “Yugoslavian coin for five pairs” What is this?

Line 149 ff The authors need to explain why they chose this above-ground reconstruction. The paper has, so far, not explained the historic descriptive record (if any) of the appearance of the towers. This needs to be far more detailed esp. as the two cited sources 9,14 are not in English

The test pits as excavated (see Figure 4) do not seem to have revealed any postholes for the foundations of the tower. Given that it could be assumed a priori that the tower would have a square foundation, the placement of area 2, as carried out by the authors, makes no sense. Surely it would have been desirable, and logical, to open up a test area that would locate two corner posts to determine the orientation and size of the tower. From an archaeological-methodological point of view the excavation strategy itself is flawed.

The list of components is very detailed. But what is the archival and historical foundation for this? The authors need to clear state what is based on evidence and what is based on interpretation/speculation. 

What is the point of providing costings for the construction in modern values? This has not been adequately explained

Line 218 to 219 what is the evidence for these items?

MINOR ISSUES

The paper needs to be edited by a native-English Speaking PROFESSIONAL editor to iron out some issues with grammar and especially expression (choice of words)